# National Knowledge-Driven Management of Obstructive Sleep Apnea—The Swedish Approach

**DOI:** 10.3390/diagnostics13061179

**Published:** 2023-03-20

**Authors:** Ludger Grote, Carl-Peter Anderberg, Danielle Friberg, Gert Grundström, Kerstin Hinz, Göran Isaksson, Tarmo Murto, Zarita Nilsson, Jonas Spaak, Göran Stillberg, Karin Söderberg, Åke Tegelberg, Jenny Theorell-Haglöw, Martin Ulander, Jan Hedner

**Affiliations:** 1Center for Sleep and Wake Disorders, Institute of Medicine, Sahlgrenska Academy, Gothenburg University, 405 30 Gothenburg, Sweden; 2Pulmonary Medicine, Sahlgrenska University Hospital, 413 45 Gothenburg, Sweden; 3Kvarterskliniken, 411 36 Gothenburg, Sweden; 4Department of Otorhinolaryngology, Surgical Sciences, Uppsala University, 752 36 Uppsala, Sweden; 5Sleep Apnea Patient Organisation (Apne Sverige), 13332 Saltsjoebaden, Sweden; 6Department for Health Care Development, Region of Västra Götaland, 40544 Gothenburg, Sweden; 7Aleris Sleep Apnea Care, 11361 Stockholm, Sweden; 8Sleep Apnea Unit, Respiratory Medicine, Umeå University Hospital, 90185 Umeå, Sweden; 9Sleep Apnea Unit, ENT Department, Ystad Hospital, 271 82 Ystad, Sweden; 10Department of Cardiology and Department of Clinical Sciences, Danderyd University Hospital, Karolinska Institute, 18288 Danderyd, Sweden; 11Sleep Apnea Unit, Capio, 70212 Örebro, Sweden; 12Sleep Apnea Patient Association (Apnefoereningen Syd), 14630 Tullinge, Sweden; 13Faculty of Odontology, Malmö University, 205 06 Malmö, Sweden; 14Department for Clinical Neurophysiology, 58185 Linköping, Sweden

**Keywords:** standardized care, treatment indication, diagnosis, CPAP, oral device, upper airway surgery, weight reduction, positional therapy

## Abstract

Introduction: This paper describes the development of “Swedish Guidelines for OSA treatment” and the underlying managed care process. The Apnea Hypopnea Index (AHI) is traditionally used as a single parameter for obstructive sleep apnea (OSA) severity classification, although poorly associated with symptomatology and outcome. We instead implement a novel matrix for shared treatment decisions based on available evidence. Methods: A national expert group including medical and dental specialists, nurses, and patient representatives developed the knowledge-driven management model. A Delphi round was performed amongst experts from all Swedish regions (N = 24). Evidence reflecting treatment effects was extracted from systematic reviews, meta-analyses, and randomized clinical trials. Results: The treatment decision in the process includes a matrix with five categories from a “very weak”” to “very strong” indication to treat, and it includes factors with potential influence on outcome, including (A) OSA-related symptoms, (B) cardiometabolic comorbidities, (C) frequency of respiratory events, and (D) age. OSA-related symptoms indicate a strong incitement to treat, whereas the absence of symptoms, age above 65 years, and no or well-controlled comorbidities indicate a weak treatment indication, irrespective of AHI. Conclusions: The novel treatment matrix is based on the effects of treatments rather than the actual frequency of respiratory events during sleep. A nationwide implementation of this matrix is ongoing, and the outcome is monitored in a prospective evaluation by means of the Swedish Sleep Apnea Registry (SESAR).

## 1. Introduction

Obstructive sleep apnea (OSA) is prevalent with up to 1 billion people worldwide [1]. OSA is characterized by repetitive partial or complete occlusion of the upper airway during sleep. The breathing events are terminated by arousal from sleep with an immediate resumption of ventilation [2]. Long-term consequences of OSA include a reduced quality of life, non-restorative sleep with increased daytime sleepiness, as well as increased risk for adverse cardiometabolic morbidities including systemic hypertension, heart failure, stroke, diabetes mellitus, and hyperlipidemia [2,3]. Untreated OSA is furthermore associated with a 2.5 fold-increased risk for traffic accidents [4].

The diagnosis of OSA is based on a careful patient history evaluating both nocturnal and daytime symptoms related to OSA. A physical examination is mandatory to assess not only risk factors for (central obesity, upper airway anatomy) but also potential consequences of OSA (e.g., high blood pressure, arrythmia, heart failure). An overnight sleep test is required to confirm frequency, duration, and physiological consequences of obstructive respiratory events during sleep.

The Apnea Hypopnea Index (AHI) is frequently used as a single parameter for obstructive sleep apnea (OSA) severity classification (mild, moderate, and severe for AHI thresholds 5- < 15, 15- < 30, and ≥30 events/hour). This classification, however, is poorly associated with patients’ symptomatology and long-term outcome [5]. To overcome limitations of the AHI-based severity classification, several new strategies have been proposed including the metric “hypoxic burden” (cumulative area under the desaturation curve) [6,7] and advanced analysis of the finger pulse wave derived from oximetry [8,9].

Treatment options are available such as positive airway pressure ventilation (PAP), mandibular advancement therapy (MAD), upper airway surgery, weight reduction therapy in overweight/obese OSA patients, and positional therapy [10]. PAP usually eliminates OSA, whereas the remaining therapies reduce the AHI by approximately 50%. The majority of patients experience good symptom control, increased quality of sleep and improved overall health-related quality of life by OSA treatment. One major limitation of PAP and MAD is reduced adherence with therapy.

A novel classification matrix, the so-called “Baveno classification” [11], divides sleep apnea patients into four classes based on OSA symptoms (“yes” vs. “no”) and comorbidities (“absence/well controlled” vs. “uncontrolled”). Distribution of patients in the four classes A–D of this classification system has been recently explored in the large European Sleep Apnea Database (ESADA) Cohort [12]. Interestingly, at least 70% of OSA patients were recommended treatment irrespective of the A–D severity classification, suggesting that the AHI alone is still the main driving factor behind a treatment decision in European sleep medicine.

A nationwide initiative to improve knowledge-driven management for health care has recently been launched in Sweden. As the political and economic responsibility for health care is decentralized, patient data from national health care quality registers are compiled to monitor regional practice within the country. The results show that medical care is significantly unevenly distributed in the 24 regions of the country. Evidence-based care may be fully available in one region, whereas only 50% of patients have access to the same therapy in other regions. This variability of both treatment tradition and patient outcome was particularly disturbing in the care of diseases with high mortality rates including cancer, acute myocardial infarction, and stroke. It was therefore decided to create national expert working groups to provide evidence-based guidelines and patient-centered standardized care plans. The implementation of these tools need to be followed up by real-time patient care data, preferably by national health care registries or by national quality registries (Figure 1). In 2016, a working group was founded by the sleep apnea quality registry (SESAR), which published national guidelines for the diagnosis of OSA [13]. Subsequently, a national expert group for obstructive sleep apnea in adults with representatives from six major health care regions in Sweden and two representatives of the national patients’ association for OSA, was officially nominated and endorsed by the regional governments. The task was to compile guidelines and a national care plan for the diagnosis and treatment of OSA in adults (www.kunskapsstyrningvard.se accessed on 15 January 2023). Despite the national aspects of care, the current manuscript describes general principles for managed care for adult patients with OSA, also potentially of clinical interest for caregivers outside Sweden.

### Aims

In the first “Swedish Guidelines for OSA treatment in adults”, we aimed to gather the current evidence for the outcomes of the different OSA treatment modalities currently available. This evidence provides the foundation for a new matrix to support caregivers and patients in the shared decision making if OSA treatment is indicated. “Whom to treat” and “How to treat”—the novel matrix—aims to provide the best available information on what treatment modality may be preferred as a first line treatment based on information gathered during the diagnostic procedure.

## 2. Materials and Methods

### 2.1. The National System for Knowledge-Driven Management within Swedish Healthcare

All major medical specialist areas created national task forces (TF) with the goal to identify disease areas with a need for national knowledge-driven management and standardized care processes. A nomination round was started for a number of diagnoses. In the TF for respiratory medicine, OSA was identified as a diagnosis with an unmet need for standardization due to long waiting lists, uneven access to care, variable use of diagnostic methods, absence of a clear competence profile for caregivers, and large regional differences in treatment traditions. Substantial regional imbalances in OSA management were detected in the SESAR registry [14]. Finally, it was decided to start a national working group for OSA in adults with the task to create both guidelines and a care plan.

#### 2.1.1. The Swedish Working Group for OSA in Adults

In 2019, the OSA working group was nominated including members from all six major health care regions in Sweden. The group included both clinical sleep professionals and patient group representatives. Several professions (i.e., physicians, dentists, and nurses) and specialties (pulmonary, internal, and general medicine, ear nose throat surgeons, specialist in neurology/neurophysiology, cardiology, sleep medicine, and dental medicine) joined the working group. The team was enforced by a process leader trained in the system of knowledge-driven management within Swedish healthcare (KS). Additional expertise was recruited from a national “guideline support group”. The work of the group reported on a regular basis to the national TF for respiratory medicine. Financial support from the Swedish Association of Local Authorities and Regions and the six major healthcare regions enabled the initiative.

#### 2.1.2. Evidence Extraction

The expert group followed a standardized process for extraction of scientific evidence. In the field of OSA, only evidence from highest scientific value described in already available systematic reviews and meta-analyses, randomized controlled trials, and large cohort studies were analyzed according to a quality ranking system.

#### 2.1.3. Delphi Round for National Consensus

Expert validation: A Delphi round was performed by distributing the draft version of the guideline amongst experts in all Swedish regions. The critical evaluation of the guideline draft was an open process, and the draft version was publicly available for review for three months. A standardized review form was provided to facilitate systematic feedback to each chapter of the guideline draft. The publication of the guideline draft is announced to all hospitals, care units, and expert groups involved in OSA management. For OSA, relevant national specialist associations for sleep medicine (SFSS), respiratory medicine (SLMF), neurophysiology (SFKNF), academy of temporomandibular disorders (SATMD), and ENT surgeons (SFOHH) informed and encouraged their members to provide feedback. Each of the 24 health care regions collected systematically feedback from their local experts in OSA care and forwarded the responses to the national working group on OSA. Altogether, 85 individuals, organizations, or official bodies provided systematic feedback during the Delphi round, and the majority of comments were included to expand the guideline from the draft to the final version. Final online publication took place in December 2021 [15].

### 2.2. The Swedish Sleep Apnea Registry (SESAR)

SESAR is one out of approximately one hundred national data quality registers in Sweden. Quality registers aim to improve medical care in the country and to reduce the large variation in care provided between regions and local health care units. Data structure and patient inclusion into SESAR have been described elsewhere in more detail [16,17]. In short, SESAR started in 2010 to register unselected patients with the diagnosis of sleep apnea. The main inclusion criteria in the quality registry is the clinical diagnosis of OSA (ICD10, G47.3) according to the International Classification of Sleep Disorders version 3(ICSD 3) criteria [18]. Sleep units voluntarily report clinical patient data derived from the diagnostic work-up into a Web-based platform. Information on therapy introduced, as well as data on treatment start and follow-up, is provided. As OSA cases are not completely captured in other Swedish registries, best possible estimates for patient coverage in the SESAR are around 50–60% for diagnosis and 85% for CPAP start based on calculation performed by the National Board of Health and Welfare. For the purpose of our analysis, we used recent data from newly diagnosed OSA patients during 2020 and 2021 [14]. SESAR data reported in this manuscript are publicly available online in the annual report from 2021 (https://sesar.registercentrum.se/om-sesar/arsrapporter/p/H1YbXyknV accessed on 10 January 2023).

## 3. Results

### 3.1. Current Status on Treatment Prescription in Sweden

During 2020 and 2021, diagnostic visits from 8529 and 11,053 patients, respectively, were reported by 29 sleep centers across the country. Female (34.2%) and male (65.8%) sleep apnea patients registered in SESAR had a mean age of 57 ± 14 and 54 ± 14 years and a mean body mass index of 32 ± 7 and 31 ± 6 kg/m^2^, respectively. At the end of the diagnostic process, the first line treatment recommendation is listed. For the years 2020–21, the most frequent recommendations were for CPAP alone (50.6%), CPAP plus weight reduction (12.9%), MAD alone (18.1%) and MAD plus weight reduction (2.5%), weight reduction alone (3.3%), and upper airway surgery alone (0.4%).

We observed a high between-center variability in treatment tradition: the mean AHI in patients recommended MAD compared with PAP therapy differed by approximately 20 units (15 vs. 35) between centers (Figure 1). In addition, the mean AHI values for each sleep center varies considerably (Figure 1). Few centers reported only patients with CPAP prescriptions. In contrast, the between-center variation in the mean Epworth Sleepiness Scale (ESS) score for patients referred for PAP or MAD treatment is low (mean ESS approximately 10 vs. 9, respectively, Figure 2). Again, there is considerable variation between the sleep units for mean ESS value per treatment option.

### 3.2. Nationwide Knowledge Management and Shared Decision Making for Treatment

The first Swedish guideline for diagnostic procedures in OSA, published in 2018, clearly stated three principal components for the diagnostic process in case of suspected OSA [13]. Component 1 performed in the dedicated sleep apnea unit is the thorough patient history including OSA-related symptoms during daytime and sleep, see Table 1. Furthermore, the occupational risk for unintentional sleep, and/or risk for sleepy driving, needs to be evaluated and documented in the medical records. Potential cardiometabolic comorbidities associated with OSA should be assessed (for examples, see Table 1). Component 2 includes an assessment of the physical status including anthropometrics, weight, length, and office blood pressure (optional). Symptoms or findings suggestive of important differential diagnoses such as central sleep apnea, nocturnal hypoventilation, as well as disorders leading to excessive daytime sleepiness are noted. ECG, pulmonary function test, and/or blood gas analysis may be indicated in selected cases. Component 3 includes the overnight sleep test performed according to predefined standards of the American Academy of Sleep Medicine (AASM) [19,20]. In Sweden, most sleep diagnostic tests are performed as Home Sleep Apnea Testing using AASM level 3 devices. Recordings are manually analyzed with support of automated scoring algorithms. Distinct national criteria have been defined for hypopnea detection (e.g., 3% desaturation criteria) and a desaturation analysis (ODI 4% criteria) [13]. National between-center differences in event scoring have been reported by the SESAR quality registry since 2013. At the end of the diagnostic process, the caregiver, often the sleep physician in charge, decides if the patient fulfills the ICSD 3 criteria for an OSA diagnosis and, as a second step, whether the respiratory disturbance during sleep is clinically relevant and may need further treatment. It is emphasized that the AHI per se is not sufficient to decide if the OSA is clinically relevant or not.

The second national document is focused solely on the treatment of OSA for adults and follows up on the decision point at the end of the diagnostic process [15]. The guideline provides a novel matrix which aims to guide both the caregiver/physician and the patient through the shared decision making for the first line treatment in newly diagnosed OSA (Figure 3). The matrix includes several layers of decision points including (A) OSA-related symptoms, (B) occurrence and state of control of cardiometabolic comorbidities, (C) frequency of respiratory events stratified for established cut-off values of 5, 15, and 30 events/hour, and (D) age with the arbitrary cut-off of 65 years. Symptoms and the most important cardiometabolic comorbidities to be taken into consideration are listed in Table 1. Subsequently, the matrix includes 5 levels of treatment indications varying from “very weak”, “weak”, “moderate”, “strong” to “very strong”.

As final step of the diagnostic work, shared decision making about the treatment options needs to be performed together with the patient (Figure 4). Evidence suggests the use of the following five treatment principles: PAP, MAD, upper airway surgery (tonsillectomy, UPPP), positional therapy, and weight reduction therapy in overweight/obese patients. The expected effect sizes on outcomes, side effects, and risks are discussed for each treatment modality. The aim is to provide standardized knowledge for the patient-centered, individualized recommendation for treatment during shared decision making. The matrix may provide better guidance both for the medical caregiver, typically the sleep expert, as well as for the patient. The current treatment guideline has been advertised publicly (professional associations, lectures at conferences, and articles in local newspapers) and for the patients by the OSA patient organizations in Sweden (annual meetings, social media, and membership magazines).

## 4. Discussion

Routines for OSA management, in particular treatment recommendations, vary substantially between centers and health care regions in Sweden, as evidenced by data from the national OSA patient registry. We propose a knowledge-driven management based on a novel treatment decision matrix, supporting both the caregiver and the patient for shared decision making addressing the first line treatment for OSA in the individual patient. The novel matrix has been calibrated with expected treatment outcomes for improvement in symptoms and comorbidities rather than frequency of respiratory events alone.

As recently outlined by an ad hoc working group of European sleep centers aligned with the ESADA-network and other academic sleep centers, clinical practice in OSA management is diverse in Europe [21]. Scoring criteria and treatment decisions vary substantially between European countries. However, compared with a similar survey 10 years before [22], a clear trend towards homogenization of procedures and an increased proportion of certified sleep medicine specialists may be observed. Indeed, the European Respiratory Society has published recommendations for the scoring of polygraphic sleep recordings [23] and the current evidence for non-CPAP therapies in OSA [24]. The European Sleep Research Society has launched a certification program for sleep medicine specialists and technicians, which has significantly contributed to the dissemination and broadening of knowledge in the field [25].

To improve congruence in the diagnostic process, the American Academy of Sleep Medicine (AASM) has published frequently updated scoring guidelines for sleep recordings in recent decades (www.aasm.org accessed on 20 January 2023), which have a world-wide impact on the scoring of sleep apnea events [5]. Studies have showed that only slight variations in the AASM hypopnea scoring rules fundamentally influences the final AHI [26]. Even more variation in standards occur when the different versions of the AASM scoring guidelines are used inconsistently within and between countries [21]. This practice has a significant impact on outcomes such as final diagnosis, the severity classification of OSA, and treatment recommendation. It is therefore recommended to decide on the national level which diagnostic criteria should be used.

There is a weak association between symptom burden and the intensity of sleep apnea frequency captured by the AHI [2]. Therefore, AHI has been taken into consideration in the matrix but to a lesser extent compared to former clinical practice [5]. Recently, measures of hypoxic load during sleep have been advocated as better predictors for risk and symptoms [6]. However, these measures of hypoxic burden are not yet fully integrated, neither in routine sleep diagnostic equipment nor in “real life” clinical work. Therefore, the panel decided to use the AHI and to wait for further guidance, standardization of calculation, and normative values for alternative parameters including “hypoxic burden”. It is therefore a major research priority to develop better parameters for the characterization of sleep apnea severity than the currently used AHI.

The Swedish treatment guideline aims to promote standardization and personalized medicine at the same time. To align the process of OSA treatment, we developed the shared decision matrix. OSA treatment, for example, with PAP, improves a variety of daytime and sleep related symptoms irrespective of patients age and AHI at baseline [27]. This implicates a strong indication for treatment (moderate to very strong) in symptomatic OSA patients, in particular, as treatment options such as PAP and MAD are safe. In addition, patients seek for treatment to improve health status and to reduce the risk for adverse cardiovascular events. OSA treatment with PAP and MAD has consistent but small effects on elevated blood pressure and metabolic parameters, which may have clinical significance for younger patients with uncontrolled cardiometabolic disease even in the absence of symptoms (moderate to strong treatment indication) [10]. In contrast, treatment indication becomes weak in asymptomatic, otherwise healthy patients with OSA, in particular in the asymptomatic elderly. The matrix aims to better translate this current evidence into clinical routine at the sleep center level. On the other hand, the guideline strongly promotes the component of individualized medicine by highlighting the need for an informed discussion between caregiver and patient regarding the pros and cons of different treatment options, as outlined in Figure 4. Conceptually, we see standardization of care and individualization as complementary, and not as contradictory elements of our care model.

Several strengths of the national knowledge-driven management process for OSA care need to be highlighted. First, the general support from all regions in the country to establish an expert group including all relevant medical specialties and professional groups involved in the OSA management process is considerable. Importantly, patient representatives participated, which generated wider insights and a true patient-centered perspective. A national Delphi round was implemented before finalization of guidelines and generated important feedback. The pre-defined process allowed a broadening of both competences and final acceptance of national guidelines through participation. A limitation is that the Swedish guideline process includes restricted resources to perform systematic literature research and meta-analyses for each treatment procedure available. In addition, the treatment matrix is not fully supported by the highest level of scientific evidence since large, multinational epidemiological studies, and large-scale randomized trials, are scarce in the field. However, we used available current and updated publications and meta-analyses performed by other societies such as the American Thoracic Society [28] and the comprehensive Canadian Health Technology Assessment for OSA management including more than 40 meta-analyses of treatment studies [10].

The new guideline has several implications for future research. The decision matrix was published in December 2021, and strategies for nationwide implementation are ongoing. In addition, prospective evaluation of alignment with the new treatment matrix and evaluation of treatment outcomes using this matrix is ongoing. The current structure in Sweden allows us to combine data from the national quality registry (SESAR) and other important national registries (cause of death, hospital diagnosis, cancer registry, traffic accident registry, use of medication, etc.), allowing us to prospectively evaluate the outcomes of the different treatment options in OSA. A corresponding cohort has already been launched for CPAP treated patients (DISCOVERY project) and will be expanded in 2023 over the entire country and with patients treated with other treatment modalities [29].

In conclusion, the current paper describes the Swedish treatment guideline of OSA created during a national knowledge-driven management process. We presented a novel treatment matrix for the shared decision making by OSA patient and caregivers. The aim of the guideline is to promote both standardization and individualized treatment in OSA. Alignment with the guideline and treatment outcomes need to be prospectively evaluated by means of unique national quality and event registries already established in Sweden.

## Figures and Tables

**Figure 1 diagnostics-13-01179-f001:**
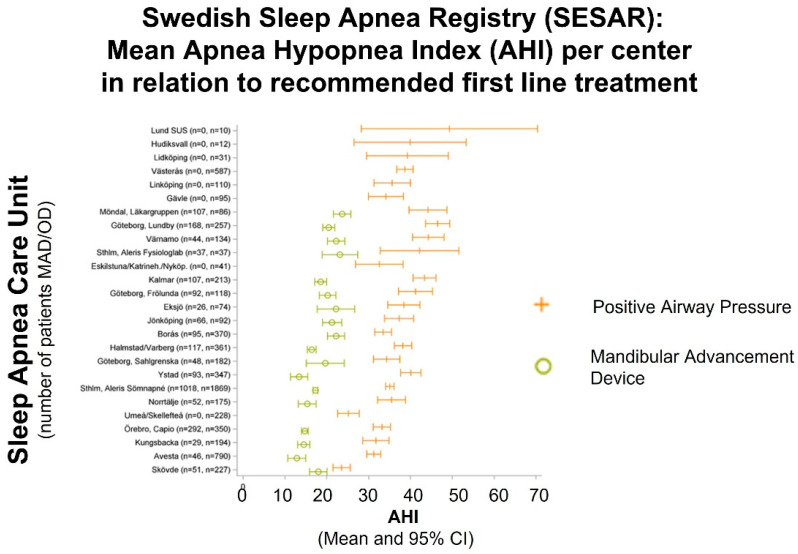
Mean Apnea Hypopnea Index (AHI) per Sleep Apnea Care Unit and primary treatment recommendation. The mean AHI is approximately 20 units lower (15 vs. 35 n/h) in patients recommended Mandibular Advancement versus Positive Airway Pressure treatment with considerable variability between centers. (Adapted from [14]).

**Figure 2 diagnostics-13-01179-f002:**
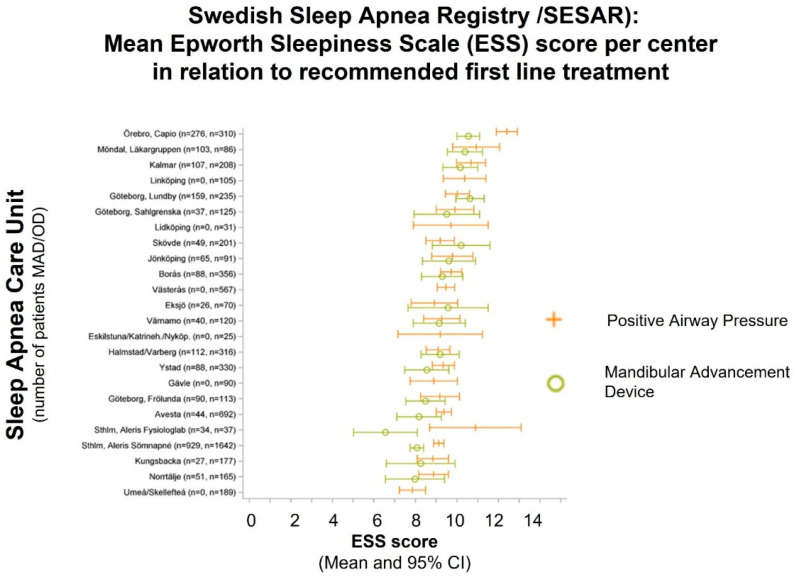
Mean Epworth Sleepiness Scale (ESS) score per Sleep Apnea Care Unit and primary treatment recommendation. The mean ESS is approximately 1 unit lower (9 vs. 10) in patients recommended Mandibular Advancement versus Positive Airway Pressure treatment, with considerable variability between centers. (Adapted from [14]).

**Figure 3 diagnostics-13-01179-f003:**
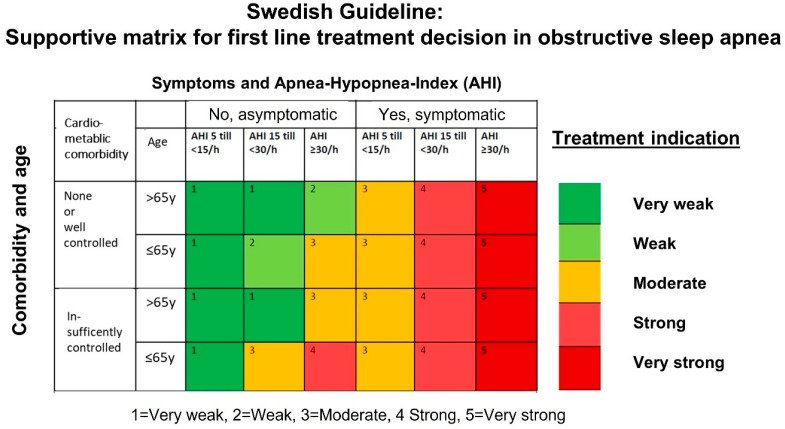
The matrix aims to provide patients and caregivers with support for the decision on treatment of patients with verified OSA. Five indication classes varying from “very weak” to “very strong” are used in combination with factors such as age, control of comorbidities, burden of symptoms, and the frequency of respiratory events in the sleep test using predefined cut-off levels. Symptom burden and relevant cardiometabolic comorbidities are listed and explained separately in the guideline and in Table 1. (Adapted from [15]).

**Figure 4 diagnostics-13-01179-f004:**
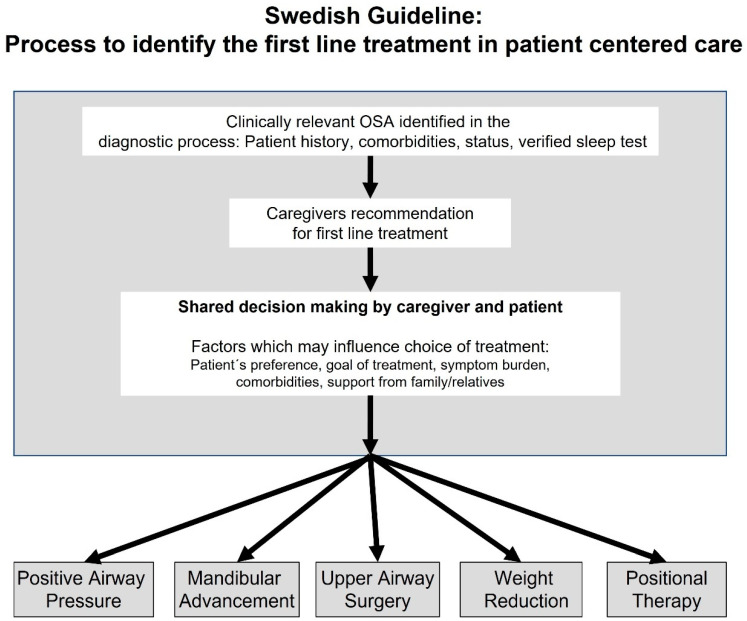
Decision tree according to the Swedish guidelines for treatment of OSA in adults. Shared decision making involving both patient and caregivers has been highlighted as an important part for an individualized treatment recommendation. (Adapted from [15]).

**Table 1 diagnostics-13-01179-t001:** Frequent OSA associated symptoms and comorbidities assessed during the diagnostic work-up.

Daytime Symptoms	Sleep Related Symptoms	Comorbidities of Interest
Daytime sleepiness	Snoring, witnessed apneas	Systemic hypertension
Non-restorative sleep	Nocturia	Atrial fibrillation
Morning headache	Nocturnal dyspnea	Ischemic heart disease
Drowsy driving	Sweating	Heart Failure
Reduced work performance	Dry mouth	Stroke
	Insomnia	Diabetes
	Nocturnal awakening	Obesity

The list of comorbidities and symptoms is incomplete but highlights the most important conditions (adapted from Hedner 2018).

## Data Availability

SESAR data are available on request when all legal requirements according to Swedish Law are fulfilled. In general, a study protocol, an ethical approval, and a data transfer agreement between institutions are required.

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
