# Peer review of "National Knowledge-Driven Management of Obstructive Sleep Apnea—The Swedish Approach"

_diagnostics, 2023, doi:10.3390/diagnostics13061179_

Round 1

Reviewer 1 Report

The title of the article is very interesting, "The Swedish Approach", but it does not report any new discoveries, only comments on future directions. It is expected that guidelines will be developed in the future that use hypoxic burden rather than AHI as an indicator.

1)In this article, authors also mentions the Swedish Guidelines for OSA treatment for adults, but no details are given. It also includes a link (https://sesar.registercentrum .se/), but it is not in English and I cannot read it. Therefore, an English version of the Swedish guidelines should be attached to the appendix.

2) Authors say that treatment indication is determined based on the matrix in Figure 3, in OSA treatment, does this mean that the patient and caregiver are free to choose the treatment method according to the shared decision tree, as presented in Figure 4? Is it possible to promote standardization of OSA treatment and personalized medicine at the same time? Wouldn't it be better to do this in stages, so that treatment is first standardized and then applied to personalized medicine?

Author Response

The title of the article is very interesting, "The Swedish Approach", but it does not report any new discoveries, only comments on future directions. It is expected that guidelines will be developed in the future that use hypoxic burden rather than AHI as an indicator.

Answer: Thank you for this general feedback. We describe a national process to improve diagnostic and therapeutic procedures in the management of OSA. One essential part of the new guidelines are the treatment decision matrix described in more detail in the manuscript. We agree with the reviewer on the hypoxic burden discussion However, there are no normal values and disease severity categories for hypoxic burden defined and validated yet. We have further emphasized the importance for other metrics than the AHI in the revised manuscript.

1)In this article, authors also mention the Swedish Guidelines for OSA treatment for adults, but no details are given. It also includes a link (https://sesar.registercentrum .se/), but it is not in English, and I cannot read it. Therefore, an English version of the Swedish guidelines should be attached to the appendix.

Answer: Thank you for the suggestion to translate the Swedish Guidelines (approximately 95 pages). However, the guideline has been compiled by national experts to adapt current evidence for different treatment modalities in OSA to the clinical settings in the Swedish Health care system. The guideline has several aspects related to the historic development of OSA care and the organizational characteristics in the Swedish health care system. Therefore, the content may not be relevant or applicable to other health care providers and we decided to write those documents in Swedish. However, to make those parts of the guideline known, which are of interest and clinical relevance for a broad audience of clinicians and researchers in the field, we decided to write the current manuscript. We highlighted the novel treatment matrix, the principles of knowledge-based care models, as well as the idea to monitor actual delivered care with the means of health care quality registries like in SESAR. We further emphasized the rationale for our paper in the introduction and discussion of the revised manuscript.

 For your perusal, we attach a machine-translated version from Swedish to English, but please note this is not a professional nor verified translation.

2) Authors say that treatment indication is determined based on the matrix in Figure 3, in OSA treatment, does this mean that the patient and caregiver are free to choose the treatment method according to the shared decision tree, as presented in Figure 4? Is it possible to promote standardization of OSA treatment and personalized medicine at the same time? Wouldn't it be better to do this in stages, so that treatment is first standardized and then applied to personalized medicine?

Answer: Thank you for this important comment. Yes, the final treatment recommendation should be based on several aspects: the frequency of apneas and/or the severity of hypoxia, comorbidities, clinical status and also patients’ preference and of course the evidence to reach the treatment goal with the suggested therapy. The standardization in this process is to present the evidence for each treatment modality for reaching a certain treatment goal, for instance to reduce AHI or blood pressure or to improve daytime sleepiness. The personalized component is embedded in the discussion with the patient – based on the clinical experience of the health care provider, the scientific evidence and the patient preference to come to the best treatment recommendation at the end of the diagnostic process. The result section of our revised manuscript was modified to clarify those important issues stated above. In the discussion we already pointed out the two aspects of standardization and individualization – we see them as complementary not contradictory. This argument has now been emphasized in the revised version of the manuscript.

Reviewer 2 Report

Dear Authors,

The manuscript National knowledge-driven management of obstructive sleep apnea – the Swedish approach is interesting. 

However, the way it is written is difficult to follow. 

It should be simplified. 

The introduction is too long. It should focus more on the aim. 

The conclusion should be more clear. 

References should be in journal style. 

Figure 3. The matrix aims to provide patients and caregivers is very important and should be further discussed. 

Figure 4 might be embodied in the manuscript. 

Author Response

Reviewer 2:

The manuscript National knowledge-driven management of obstructive sleep apnea – the Swedish approach is interesting. 

Answer: Thank you for the feedback.

However, the way it is written is difficult to follow. It should be simplified. 

Answer: In the revised manuscript we have focused more on the aim of the study and reduced the volume, to make the text more easily read. Please notice also our answer to reviewer 1 above.

The introduction is too long. It should focus more on the aim. 

Answer: We have substantially reduced the introduction in the revised manuscript and expanded on the aim of our study.

The conclusion should be more clear. 

Answer: We rewrote and clarified the conclusions in the revised manuscript.

References should be in journal style. 

Answer: We modified the references according to journal style in the revised manuscript.

Figure 3. The matrix aims to provide patients and caregivers is very important and should be further discussed. 

Answer: We further discussed figure 3 in the revised manuscript.

Figure 4 might be embodied in the manuscript. 

Answer: We further integrated figure 4 in the content of the manuscript (result section and discussion).

Round 2

Reviewer 2 Report

Congratulations on your work!